

# Little Conductor
## Projekt i implementacja do zarządzania i synchronicznego udostępniania zasobów muzycznych na grupie urządzeń



**Autorzy**: Jan Bogucki[ID] · Jakub Cebula[ID] · Mateusz Chodyń[ID] · Jan Staręga[ID]

**Opiekun:** Dr inż. Maciej Walczyński

**Streszczenie**

Projekt ten ma na celu stworzenie zaawansowanego systemu, który umożliwia zarządzanie oraz synchroniczne udostępnianie zasobów muzycznych na grupie urządzeń. W obecnych czasach, kiedy muzyka odgrywa kluczową rolę w codziennym życiu, istnieje zapotrzebowanie na rozwiązania pozwalające na łatwe i efektywne zarządzanie oraz dzielenie się muzyką pomiędzy różnymi urządzeniami, takimi jak smartfony, tablety, komputery i inteligentne głośniki. Oprócz plików audio, system będzie obsługiwał także pliki PDF z nutami oraz oferował funkcję metronomu.

# 1 TREŚĆ WŁAŚCIWA

## 1.1 Wstęp

Po zapoznaniu się z rynkiem muzycznym oraz uzyskaniu informacji o zapotrzebowaniach aktywnych muzyków, zaobserwowaliśmy że potrzebna jest aplikacja upraszczająca organizację materiałów muzycznych oraz komunikację podczas koncertu. Problemy z materiałami muzycznymi i komunikowanie zmian podczas koncertu to nieodłączne elementy życia muzyków, które generują niepotrzebny stres i rozpraszają uwagę. Nasza aplikacja ma na celu wyeliminowanie tych bolączek i ułatwienie muzykom skupienia się na grze.

Celem było stworzenie aplikacji pozwalającej dyrygentom na przechowywanie i udostępnianie odpowiednich plików pdf z nutami do każdego członka zespołu i upewnienie się, że każdy z nich otrzyma zapis nutowy właściwy dla swojego instrumentu. Ułatwia to przeprowadzanie prób oraz występów bez potrzeby żmudnego przygotowywania zestawu nut dla każdego instrumentu od nowa. Pozwala to na większe skupienie się na samej próbie lub występie.

Postawione przed zespołem zadanie miało na celu utworzyć produkt ze ścisłą integracją komponentu mobilnego z serwerowym, co umożliwia korzystanie z produktu na wielu urządzeniach, komunikację pomiędzy muzykami oraz skupienie na zespołach zamiast jednostek. Dyrygenci i liderzy zespołów powinni mieć możliwość zarządzania nimi również w naszym systemie w tym na dodawanie, pobieranie i usuwanie plików z nutami oraz zarządzanie koncertem na żywo.

## 1.2 Prace powiązane

- Najpopularniejsze aplikacje z kategorii systemów nut dla zespołów dostępne na rynku są skierowane w znacznej większości do indywidualnych użytkowników, lub edycji nut poza trwaniem koncertu.

- Najbardziej rozpoznawalną jest płatna aplikacja ForScore. Jej ograniczenia to przeznaczenie głównie do przechowywania, organizowania i anotowania nut, bez synchronizacji. Jest też wyłącznie na system iOS.

- Aplikacją z funkcjami najbardziej zbliżonymi do naszych jest Newzik. Jest to płatny, rozbudowany program z wieloma funkcjami zarządzania nutami, jednak aby otrzymać funkcje synchronizacji zbliżone do naszej aplikacji, należy wykupić subskrypcję premium (50 dolarów/rok).

- Opcje bezpłatne takie jak aplikacja Piascore i podobne również są w zdecydowanej większości organizatorami i edytorami nut, bez synchronizacji w zespole.

## 1.3 Rezultaty

**Zarządzanie użytkownikiem i grupami**

- Aby ułatwić dyrygentom kontrolowanie jednej lub wielu grup muzyków, często w małym okresie czasowym, wprowadziliśmy grupy łączące ich z muzykami w jedną jednostkę organizacyjną. Ma to na celu ułatwienie kontroli nad zasobami dla użytkownika, oraz udostępnienie możliwości wspólnego wyświetlania nut podczas występu. Dyrygent może udostępniać innym muzykom grupę za pomocą wygenerowania odpowiedniego kodu dostępu, przypisanego grupie. Aby uniknąć problemów z niezautoryzowanym dostępem może zostać on później usunięty, a nieproszeni użytkownicy usunięci przez dyrygenta w miarę potrzeby.

**Zarządzanie plikami**

- Administrator grupy (nazywany dalej dyrygentem) może udostępnić pliki pdf z nutami, określić ich metadane i umieścić na serwerze. Dodatkowo ma możliwość rozróżniania plików w zależności od instrumentu docelowego, dzięki czemu każdy muzyk w jego grupie może pobierać pliki dopasowane dla jego instrumentu.

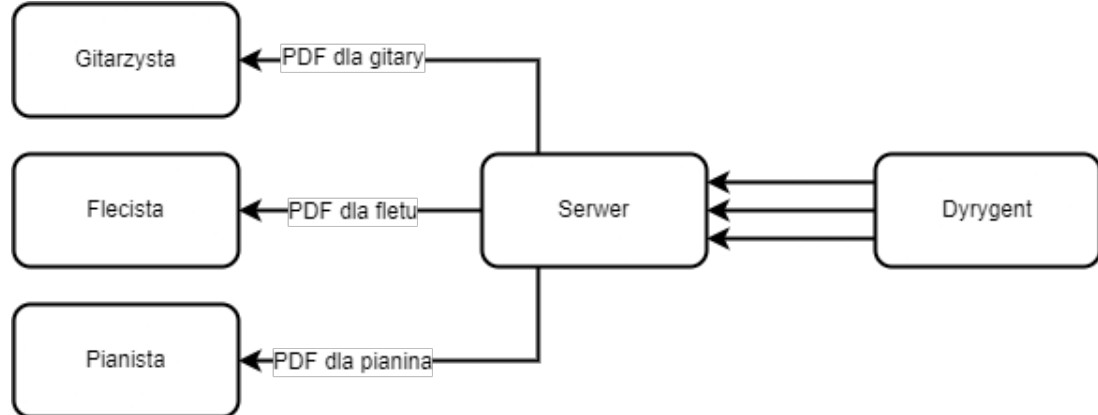

Rysunek 1: Schemat przesyłania plików

- Dyrygent może również utworzyć setlistę, czyli listę utworów w kolejności chronologicznej dla planowanego koncertu. Dzięki temu w momencie gry nie będzie musiał szukać i ręczne wyświetlać kolejnych plików z nutami, ponieważ system zrobi to za niego.

**Komunikacja w czasie rzeczywistym**

- Dzięki możliwości oznaczenia grupy za „aktywną", członkowie zespołu są w stanie przygotować się do próby lub występu. Dzięki aktywacji wybranej grupy będą oni w stanie otrzymać sygnał do otwarcia nut dla danego utworu, wysłany przez dyrygenta. Utwory można łączyć w setlisty które odpowiednio otwierają następny plik w czasie rzeczywistym.

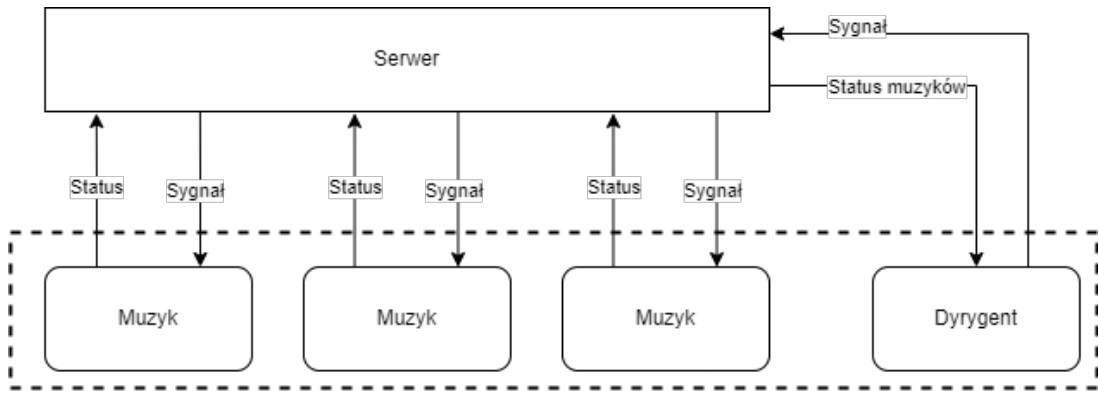

Rysunek 2: Schemat komunikacji na żywo

**Spełnione cele**

- Aplikacja umożliwia przechowywanie plików pdf na serwerze, zarządzanie plikami oraz dostępem do nich. Muzycy mogą komunikować się na żywo podczas występu, korzystając ze skutecznego systemu zarządzania danymi użytkownika, który jest zorientowany na zespoły muzyczne. Ponadto, umożliwia łatwe dołączanie oraz usuwanie użytkowników z zespołu, co przyczynia się do efektywniejszego zarządzania grupami muzycznymi.

**Wyniki testów**

- Program został przetestowany przez muzyków.

- Program działa na każdym dostępnym nam urządzeniu, poza linuxem, gdzie niefortunnie biblioteka obsługująca odczyt pdf nie jest dostępna.

## 2 WNIOSKI

W ramach projektu osiągnięto szereg kluczowych celów, które mają istotne znaczenie dla odbiorców, takich jak dyrygenci i zespoły muzyczne. Główne rezultaty obejmują stworzenie zaawansowanego systemu do zarządzania plikami PDF z nutami, synchronizacji działań w grupie oraz komunikacji w czasie rzeczywistym. Wprowadzenie grup organizacyjnych dla dyrygentów i muzyków znacząco usprawniło zarządzanie próbami oraz występami, eliminując wiele problemów związanych z tradycyjnym zarządzaniem nutami i synchronizacją zespołu. Przeprowadzono również testy z udziałem muzyków, co potwierdziło praktyczność i funkcjonalność aplikacji.

Największym sukcesem projektu jest skuteczne połączenie aplikacji mobilnej z komponentem serwerowym, co umożliwia łatwe przechowywanie danych i ich synchronizację pomiędzy różnymi urządzeniami. Funkcjonalności takie jak udostępnianie grupowe, możliwość tworzenia setlist oraz automatyczne otwieranie nut podczas koncertu podniosły komfort pracy dyrygentów i muzyków.

Projekt udowodnił swoją wartość również pod kątem technologicznym. System został zaprojektowany w sposób, który zapewnia intuicyjność obsługi oraz niezawodność. Testy użytkowników wykazały wysoką skuteczność działania aplikacji na większości platform, co czyni go dostępniejszym niż niektóre konkurencyjne rozwiązania, takie jak ForScore czy Newzik. Ponadto projekt wypełnia lukę na rynku, oferując darmowe rozwiązanie z kluczowymi funkcjonalnościami synchronizacji dla zespołów.

### 2.1 Znaczenie projektu

- **Dla użytkowników indywidualnych i zespołów:** Dzięki zaawansowanym funkcjom zarządzania zasobami muzycznymi projekt wspiera zarówno małe zespoły, jak i większe orkiestry w efektywnym przygotowywaniu prób i koncertów.

- **Dla technologii muzycznej:** Projekt przyczynia się do rozwoju narzędzi dedykowanych dyrygentom, skupiając się na automatyzacji i synchronizacji, które są kluczowe w kontekście współczesnych potrzeb muzycznych.

### 2.2 Główne osiągnięcia

- **Przechowywanie i synchronizacja plików PDF na serwerze:** Umożliwiono łatwe udostępnianie plików z nutami dla wszystkich członków zespołu w sposób szybki i zorganizowany.

- **Zarządzanie zespołami i użytkownikami:** Stworzono system, który pozwala na łatwe dołączanie oraz usuwanie członków zespołu, co znacząco ułatwia zarządzanie grupą.

- **Komunikacja na żywo:** Wprowadzono możliwość synchronizacji otwierania plików nutowych w czasie rzeczywistym, co eliminuje potrzebę manualnego wyszukiwania utworów podczas występów.

- **Intuicyjne interfejsy i narzędzia:** System umożliwia dyrygentom tworzenie setlist oraz ich dynamiczne zarządzanie w trakcie występów.

### 2.3 Wnioski technologiczne

Testy wykazały, że aplikacja działa poprawnie na większości systemów operacyjnych, z wyjątkiem systemów Linux, gdzie ograniczeniem jest brak dostępności kluczowej biblioteki do obsługi plików PDF. Niemniej jednak, aplikacja wykazała wysoką stabilność i łatwość integracji w środowisku produkcyjnym.

## 2.4 Podsumowanie

Projekt spełnia wszystkie założone cele i wyznacza nowe standardy w zakresie zarządzania zasobami muzycznymi, oferując skalowalne i efektywne narzędzie dla dyrygentów i muzyków. Przyszły rozwój produktu może dodatkowo zwiększyć jego zasięg oraz funkcjonalność, co czyni go obiecującym rozwiązaniem dla globalnego rynku muzycznego.

## 2.5 Przyszłe kierunki rozwoju

· W przyszłości planowany jest rozwój w kierunku zwiększenia skalowalności. Możliwe rozwiązania to np. zastosowanie Ribbon w celu rozłożenia obciążenia lub podzielenie serwerów na odpowiadające im regiony w świecie rzeczywistym. Następnie należałoby wprowadzić regularną synchronizację z jednym, lub wieloma serwerami w celu uzyskania zabezpieczonej kopii danych użytkowników i usług w różnych częściach świata.

· W ramach dodatkowych funkcjonalności planowane jest dodanie możliwości umieszczania notatek na plikach nutowych w formie tekstu lub nawet zwykłego rysowania. Następnym pomysłem jest wprowadzenie serwera lokalnego łączącego się z użytkownikami w sieci lokalnej. Przechowujący pliki bez potrzeby posiadania połączenia z Internetem.

## 2.6 Podziękowania

· Podziękowania dla Pana Doktora Walczyńskiego i Pana Doktora Kopla za wysoki merytoryczny wkład w stworzenie produktu. W szczególności dziękujemy za wysoki poziom znajomości tematu w formie wymagań, jakie może postawić normalny zespół muzyczny w celu przeprowadzenia udanego występu.

## 2.7 Bibliografia

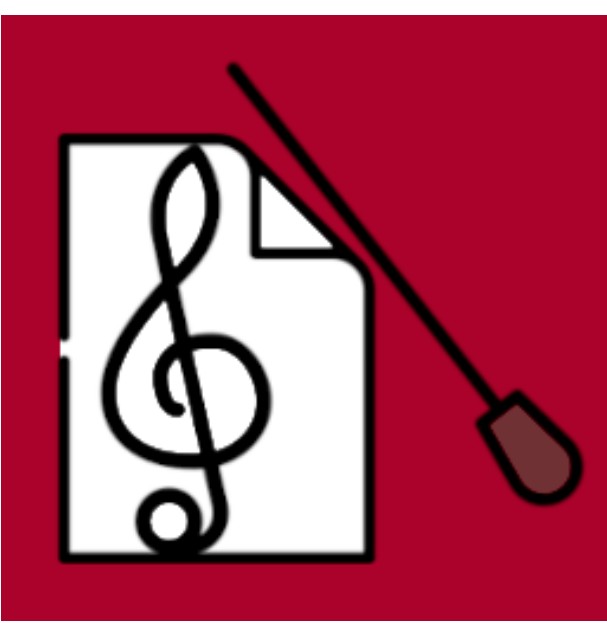

Rysunek 3: To jest logo naszej aplikacji

· Dokumentacja Fluttera [1].

· Dokumentacja Go [4].

· Google authentication in GO [3]

· What is the Go [2].

# LITERATURA

[1] Google. *Flutter Cookbook.* Google, May 12, 2017.

[2] Alexander S. Gillis Nick Barney. *What is the Go or Golang programming language?* www.techtarget.com, February 2023.

[3] Permify. *How to Implement OAuth 2.0 into a Golang App.* Permify.co - All rights reserved, May 30, 2024.

[4] Rob Pike Robert Griesemer and Ken Thompson. *Dokumentacja GO.* Google, 2009.