# OpenReview forum: "Little Conductor - Projekt i implementacja do zarządzania i synchronicznego udostępniania zasobów muzycznych na grupie urządzeń"
_pwr.edu.pl/Wrocław_University_of_Science_and_Technology/2024/ZPI_Day — Wrocław University of Science and Technology 2024 ZPI Day Submission_

### Official Review · Reviewer_wXBM · 2024-12-03
**Well tested, but badly written**

**Confidence:** 4
**Significance Of Results:** 4
**Overall Quality:** 3

**Compliance With Template:**

3: Average Quality – The article includes most of the required sections, but some may be incomplete, written in a general or unclear manner. The content is correct but requires further refinement.

**Description Of Results:**

3: Average Quality – The results are described with moderate detail. Some examples or evaluation elements are present but insufficiently developed or incomplete.

**Feedback On Consistency:**

The description mixes the order, jumping between sections. The language is in places highly unprofessional, most visible in the Figure description. Otherwise all the sections are included, just not in logical order.

**Potential For Development:**

The authors point potential problems with scalability in future development, but would it really be one - largest orchestras are something ca. 100-200 people (?), so the scale is limited. I would instead focus on UX development (and initially - testing).

**Project Nature Evaluation:**

This is a well done engineering work. I am especially happy that it was tested with real potential users - musicians. It has high potential for utility as a deployed application. The description of technical side could be improved slightly and the test results given publicly.

**Technical Language Precision:**

2: Low Quality – The language is partially inappropriate. Significant terminology errors and numerous ambiguities are present. Some sections are imprecise or inconsistent with the expected style of a technical report.

---

### Official Review · Reviewer_47Me · 2024-12-04
**W pracy - której jestem "promotorem" - Autorzy skupili się na współpracy z zespołem jazzowym zgłaszającym zapotrzebowanie na wyspecjalizowaną aplikację pozwalającą na zaawansowane i efektywne zarządzanie rozbudowaną biblioteką nut. Rozwiązanie to może zostać przyjęte przez tzw. rynek muzyczny, ponieważ brakuje wciąć tego typu rozwiązań, które pozwalałyby na spełnienie tego niewielkiego i mocno wyspecjalizowanego rynku.**

**Confidence:** 5
**Significance Of Results:** 5
**Overall Quality:** 4

**Compliance With Template:**

5: Very High Quality – The article contains all the required sections, which are written in a very detailed, clear, and error-free manner. The structure is professional and meets expectations, and the content adheres to the highest substantive and formal standards.

**Description Of Results:**

4: High Quality – The results are described in detail and supported by usage examples or evaluations. The description is reliable but may lack full depth of analysis.

**Feedback On Consistency:**

W pracy - której jestem "promotorem" - Autorzy skupili się na współpracy z zespołem jazzowym zgłaszającym zapotrzebowanie na wyspecjalizowaną aplikację pozwalającą na zaawansowane i efektywne zarządzanie rozbudowaną biblioteką nut. Rozwiązanie to może zostać przyjęte przez tzw. rynek muzyczny, ponieważ brakuje wciąż tego typu rozwiązań, które pozwalałyby na spełnienie tego niewielkiego i mocno wyspecjalizowanego rynku.
Opis projektu jest zgodny z założeniami, które ograniczają go do około 4-ch stron. Jednak wydaje mi się, że mógłby być rozszerzony, aby Czytelnik mógł się z opisu dowiedzieć o wszystkich aspektach prezentowanego rozwiązania.

**Potential For Development:**

Projekt może być wdrożony w pracy zespołu jazzowego. Ponadto może być upubliczniony, jako ogólnodostępna aplikacja z możliwością indywidualnego dostosowania potrzeb potencjalnych klientów.

**Project Nature Evaluation:**

Projekt ma typowo inżynierski charakter. Wykorzystane narzędzia i metody są dobrane i zastosowane w przemyślany i prawidłowy sposób. Rezultat - ze względu na swoje pierwotne założenia skierowane do wąskiej grupy odbiorców - w postaci wytworzonej aplikacji spełnia jednak wszystkie postulaty (wręcz z nawiązką, gdyż w trakcie pracy Autorzy po konsultacji z zespołem dodali funkcjonalność czteromiarowego metronomu)

**Technical Language Precision:**

5: Very High Quality – The language is entirely appropriate for a technical report. All terms are used correctly and precisely, and the style is professional, clear, and coherent, without any errors or ambiguities.

---

### Official Review · Reviewer_GXUk · 2024-12-06
**Recenzja Little Conductor**

**Confidence:** 4
**Significance Of Results:** 4
**Overall Quality:** 4

**Compliance With Template:**

4: High Quality – The article contains all the required sections, which are well-written and substantively correct, although minor errors or shortcomings may be present. The overall structure is clear and coherent.

**Description Of Results:**

4: High Quality – The results are described in detail and supported by usage examples or evaluations. The description is reliable but may lack full depth of analysis.

**Feedback On Consistency:**

Analiza problemu, cel i zakres funkcjonalny sposób został przedstawiony w zwięzły i jasny sposób. Prezentacja wyników jest przedstawiona w kontekście zdefiniowanych problemów a także umotywowana przykładami użycia. Podsumowanie i proponowana dalsza praca na projektem wydaje się być logicznym następstwem uzyskanych rezultatów.

**Potential For Development:**

W pracy omówiono potencjał rozwoju projektu z perspektywy technicznej, jak w minimalnym zakresie w zakresie funkcjonalnym.

**Project Nature Evaluation:**

Niestety w pracy minimalnie poruszono temat użytych technologi, procesu wytwórczego, a przede wszystkim nie uzasadniono w żaden sposób jej wyboru, słabo także przedstawiono projekt/architekturę systemu.

**Technical Language Precision:**

2: Low Quality – The language is partially inappropriate. Significant terminology errors and numerous ambiguities are present. Some sections are imprecise or inconsistent with the expected style of a technical report.

---

### Decision · Program_Chairs · 2024-12-10

Accept (Poster)